# ALGD-ORB: An improved image feature extraction algorithm with adaptive threshold and local gray difference

**Guoming Chu** *, **Yan Peng, Xuhong Luo**

School of Automation and Information Engineering, Sichuan University of Science and Engineering, Yibin, China

* 759053364@qq.com

## Abstract

Simultaneous Localization and Mapping (SLAM) technology is crucial for achieving spatial localization and autonomous navigation. Finding image features that are representative presents a key challenge in visual SLAM systems. The widely used ORB (Oriented FAST and Rotating BRIEF) algorithm achieves rapid image feature extraction. However, traditional ORB algorithms face issues such as dense, overlapping feature points, and imbalanced distribution, resulting in mismatches and redundancies. This paper introduces an image feature extraction algorithm called Adaptive Threshold and Local Gray Difference-ORB(ALGD-ORB) to address these limitations. Specifically, an adaptive threshold is employed to enhance feature point detection, and an improved quadtree method is used to homogenize feature point distribution. This method combines feature descriptors generated from both gray size and gray difference to enhance feature descriptor distinctiveness. By fusing these descriptors, their effectiveness is improved. Experimental results demonstrate that the ALGD-ORB algorithm significantly enhances the uniformity of feature point distribution compared to other algorithms, while maintaining accuracy and real-time performance.

## Introduction

### Background

Simultaneous Localization and Mapping (SLAM) represents the forefront of spatial localization technology and has exhibited remarkable progress over the last three decades. SLAM involves the process in which a robot or vehicle equipped with a specific sensor constructs a model of its surrounding environment while in motion, without prior knowledge of the environment. It simultaneously estimates its own movement, enabling autonomous positioning and navigation. Visual SLAM (VSLAM) has emerged as a prominent area of research due to its cost-effectiveness, ability to capture abundant information, and ease of installation, attracting substantial attention in the research community [1]. Finding representative image features from images is a key research problem for visual SLAM [2]. The extraction and matching of image feature points play a crucial role in various tasks, including image retrieval, object tracking, 3D reconstruction, image registration, pose estimation, map creation, and loop detection.

**Data Availability Statement:** The dataset is publicly available and can be accessed at Oxford Dataset (https://www.robots.ox.ac.uk/~vgg/research/affine/).

**Funding:** Key Laboratory of Enterprise Informationization and Internet of Things Measurement and Control Technology in Sichuan Province Universities: 2021WYJ04 The funders had no role in study design, data collection and analysis, decision to publish, or preparation of the manuscript.

**Competing interests:** The authors have declared that no competing interests exist.

The significance of feature point detection-based image matching methods has been increasingly emphasized with the advancement of visual SLAM technology [3, 4].

## Current challenges or limits of prior SOTAs

Historically, researchers have developed many prominent feature extractors and descriptors, each of which exhibits different performance in terms of rotation and scale invariance as well as speed [5]. David Lowe introduced the Scale Invariant Feature Transform (SIFT [6] method in 1999, and its subsequent refinement in 2004 has established it as a highly renowned and influential feature extraction algorithm. SIFT features are local image descriptors that maintain robustness against a wide range of transformations, including rotation, scale changes, viewpoint changes, lighting variations, affine transformations, and noise. However, SIFT features obtain the local extreme values of the space through a pixel-by-pixel traversal in DOG (Gaussian First-order Difference) space. Even though SIFT features perform well in most cases of transformation, its large computational cost leads to low extraction efficiency, which seriously limits its application in real-time systems. Meanwhile, the effectiveness of SIFT may also be affected in images with strong noise or complex background. Herbert Bay proposed the Speeded-Up Robust Features (SURF) [7] algorithm in 2006 by referring to the SIFT algorithm. Compared with SIFT, the algorithm uses Hessian matrix and dimensionality reduction feature descriptors, which improves the execution efficiency to some extent. SURF overcomes the limitations of SIFT in working in high-dimensional environments and is also robust to lighting changes. However, it is important to note that this advantage comes with a trade-off, as it has lower performance in terms of resource consumption. Additionally, while SURF has improved efficiency, it is still limited for real-time applications due to its computational complexity. This method was applied to the first monocular visual SLAM (Mono SLAM) [8] system proposed by Davison et al. in 2007. In 2006, Edward Rosten first proposed the FAST (the Features from Accelerated Segment Test) algorithm [9], which uses the gray of each pixel in a particular region and the gray of the center to determine whether the central point is a corner, greatly improving speed. Due to the fact that the algorithm only detects corner points, it does not have scale invariant and rotation invariant. Compared with SIFT and SURF algorithms, this algorithm is susceptible to noise and has lower accuracy.

Binary Robust Independent Elementary Features (BRIEF) [10, 11] is a description of the detected feature points, which is a binary encoded descriptor, and these comparisons produce a binary string that can be matched very quickly using the exclusive OR (XOR) operation. BRIEF is constructed using comparisons of the intensities of random pixel pairings within a patch centered on the feature point and discards the traditional method of describing feature points using a regional gray histogram, so it is much faster than SIFT-like descriptor extraction [12]. In the field of computer vision, the research community has introduced numerous binary descriptor algorithms in recent years, aiming to enhance the performance of image matching algorithms. Binary descriptors leverage efficient bit operations to expedite the process of feature point matching. Compared with non-binary descriptors, binary descriptors offer significant improvements in computational efficiency and storage space. The Oriented FAST and Rotated BRIEF (ORB) matching algorithm, initially proposed by Rublee et al. [13], is a fusion of the FAST feature detection algorithm and the enhanced BRIEF feature description algorithm. The ORB algorithm generates binary descriptors by employing machine learning-based selection of point pairs, while incorporating rotational invariance by calculating the corner points' orientation using the gray centroid. It is about 100 times faster than SIFT algorithm and 10 times faster than SURF algorithm [13]. Traditional ORB algorithms, however, do not provide good matching quality and anti-interference capabilities.

The advantage of the ORB algorithm is the fast computation speed, however, the stability of the ORB algorithm is weaker than SIFT descriptor and SURF descriptor, but considering that the visual SLAM algorithm needs high real-time performance, ORB feature points are more suitable for feature point SLAM algorithm. Since the ORB algorithm pursues the speed advantage too much, the ORB algorithm does not work well on scale invariance, illumination invariance, and rotational invariance, which will lead to a large number of mismatches. Numerous enhanced image-matching algorithms have been proposed based on the ORB algorithm. For instance, BRISK (Binary Robust Invariant Scalable Keypoints) [14] is a feature detection and description algorithm introduced by Leutenegger et al. at the 2011 ICCV, which is known for its robustness to changes in scale, rotation, and viewpoint, making it suitable for various image matching and recognition tasks. The FREAK algorithm [15] was presented at the 2012 International Conference on Computer Vision and Pattern Recognition, which is based on the BRISK algorithm with the introduction of feature descriptors in binary. The FREAK algorithm not only has the characteristics of high matching rate and high operation rate than the traditional detection algorithm, but also has the scale transformation invariant property of stronger stability and stronger robustness. The feature descriptors of the FREAK algorithm are made up of the central feature point and the sampled feature points combined. In 2015, according to Yang and Cheng [16], the Local Difference Binary (LDB) algorithm is proposed as an ultrafast and distinctive feature description method that enhances traditional binary descriptors by incorporating gradient information in both horizontal and vertical directions. LDB partitions feature point regions into different-sized sub-blocks to capture local information, enabling smaller subblocks to gather localized information for improved descriptor discrimination. Conversely, larger subblocks effectively suppress noise but may result in reduced distinctiveness due to decreased sensitivity to subtle changes. In 2017, Levi et al. proposed Learned Arrangements of Three Patch Codes (LATCH) [17]. LATCH improves upon the original local binary feature description by utilizing multi-image block F norm comparison instead of pixel-level gray level comparisons employed by ORB. This modification enhances the algorithm's robustness against noise, contributing to improved anti-noise performance.

Traditional sparse feature extraction methods typically adopt a "detect-then-describe" strategy, where keypoint detectors [6, 7, 18–20] are used to extract keypoints, and feature descriptors [6, 7, 10, 13, 14] are subsequently extracted from these keypoints. However, in recent years, deep learning-based methods have gained attention, and some of these methods integrate the learning of descriptors and detectors. LIFT (Learned Invariant Feature Transform) [21] is the first fully learning-based architecture that reconstructs the key steps of the SIFT algorithm using neural networks to achieve stable feature extraction. SuperPoint [22], inspired by LIFT, treats keypoint detection as a supervised task. In the initial stage, it trains the model using labeled synthetic data to perform keypoint detection before description. Later, SuperPoint further extends to an unsupervised version [23]. DELF (Deep Local Features) [24] and LF-Net (Local Features Network) [25] employ distinct methods for unsupervised learning, with DELF utilizing attention mechanisms, while LF-Net adopts an asymmetric gradient back-propagation scheme. D2-Net [26], unlike previous approaches that separately learn descriptors and detectors, devises a joint optimization framework based on non-maximum suppression, allowing simultaneous optimization of descriptor and detector performance. To enhance the reliability and repeatability of keypoints, R2D2 [27] introduces a list-wise sorting loss function based on differentiable mean average precision to improve keypoint ranking. Similarly, aiming for the same objective, ASLFeat [28] introduces deformable convolutions to enhance the reliability and repeatability of keypoints. F. Luo et al. [29] propose a multiscale diff-changed feature fusion network (MSDFFN) for change detection (CD) in hyperspectral images (HSI). This network enhances the feature representation by learning fine-grained change

components between dual-temporal HSI at different scales. In 2013, Wang et al. [30] introduced FeatureBooster, a lightweight network designed to enhance keypoint descriptors within the same image. Utilizing both MLP-based (Multi-layer perceptron) and Transformer-based stages, the network boosts the performance of both hand-crafted and learning-based descriptors. Wang and Liu [31] developed SSG-Net, an innovative approach for registering multimodal images. This technique incorporates an adaptive training framework and initially employs both the SuperPoint and SuperGlue networks for coarse alignment. Subsequently, the DEGENSAC algorithm is applied to identify the best registration points, using a judiciously chosen threshold.

In 2017, the quadtree approach was employed by ORB SLAM 2 for the homogenization of ORB feature extraction [32, 33]. While this strategy undeniably enhances the uniformity of feature point distribution, there is still room for improvement in terms of efficiency. Building upon this approach, YU Xinyi et al. [34] proposed the Qtree-ORB algorithm in 2018, which effectively eliminates redundant feature points. However, this method still relies on the traditional quadtree structure, necessitating further enhancements to improve computational efficiency. In order to extract additional feature points, Xin-Nan et al. [35] adopted adaptive thresholding in 2019 as opposed to fixed thresholding in the original ORB method. Nonetheless, the aforementioned approach requires the computation of a threshold value on a per-pixel basis, leading to increased computation time. Ma et al. [36] presented a feature description technique aimed at enhancing the accuracy of the ORB algorithm in 2020. Their method incorporates local grayscale difference information alongside the original grayscale intensity information. In 2013, Zhu et al. [37] developed an improved ORB-RANSAC algorithm tailored for Unmanned Aerial Vehicle (UAV) remote sensing image registration. The method employs a simplified image pyramid for scale invariance and an adaptive FAST corner detection algorithm. It also utilizes bidirectional matching and an enhanced RANSAC for fine feature point matching.

While traditional algorithms like SIFT and SURF have been foundational in the field, they often suffer from computational inefficiency and lack of robustness in varying conditions. Deep learning-based methods like LIFT and SuperPoint have shown promise in terms of performance but require large amounts of labeled data and computational resources. The integration of these methods into existing SLAM systems remains an open challenge.

## Research motivation

The motivation behind this research is centered around addressing certain limitations associated with the ORB (Oriented FAST and Rotated BRIEF) algorithm's feature point detection and description techniques. Through a comprehensive analysis of the ORB algorithm, several noteworthy shortcomings have been identified:

1. The ORB algorithm incorporates the FAST method for feature point detection, which examines the gray value in the vicinity of each candidate feature point. By analyzing the pixel values within a circular region centered on the candidate point, the algorithm determines if there are sufficient pixel points exhibiting significant deviations from the grayscale intensity of the candidate point. The FAST method, on the other hand, employs a randomly chosen globally fixed threshold, which results in the identified feature points being too concentrated or even overlapped. In regions where the picture texture is not visible, this may result in the extraction of only a limited number of feature points or none at all. Such dense distributions of feature points can significantly impact system accuracy in various scenarios, including image navigation, video tracking, and SLAM systems [38].

2. The ORB algorithm employs the enhanced BRIEF algorithm to compute feature point descriptors. The BRIEF algorithm generates binary descriptors by comparing the intensities of two points within a window centered on the feature point. However, this process solely relies on the size relationship between gray values, thereby neglecting the utilization of valuable image information contained in the pixel intensity differences. This limitation leads to a potential loss of image information.

Addressing these limitations is of paramount significance in bolstering the precision and performance of the ORB algorithm within image matching. This enhancement in capabilities renders the algorithm more adept at tasks like image navigation, video tracking, and SLAM systems. The core objective of this research is to architect strategic measures that effectively mitigate these challenges, thereby contributing substantially to the advancement of cutting-edge techniques in feature point detection and description.

## Main contributions

This paper aims to address the aforementioned limitations of the ORB algorithm. The main contributions of this study can be summarized as follows:

1. In the ORB SLAM system proposed by Mur-Artal [32, 33], a fixed two-threshold approach was used to detect FAST feature points. A higher threshold of 20 was first applied for feature point detection. If no points were detected with this threshold, a lower threshold of 7 was then used. However, this dual-thresholding method relied on preset fixed threshold values without considering variations in local pixel intensities. As a result, it may lack robustness under changing lighting conditions compared to an adaptive thresholding approach. To address this limitation, we propose an adaptive thresholding method based on the analysis of image grayscale contrast. This enhancement improves the feature point extraction capability of the original ORB algorithm, especially in regions with weak textures.

2. In the work by Mur-Arta [32, 33], a quadtree approach was employed to address the issues of over concentration and overlapping of feature points. However, the quadtree splitting process is based on the total number of feature points extracted from each pyramid image layer. This approach may lead to a decrease in computational efficiency, as it requires further sorting and splitting of nodes when the total number of feature points can still be satisfied. To improve the efficiency of feature extraction, our proposed method reduces the estimation of redundant feature points while enhancing the overall feature extraction efficiency.

3. During the feature point description stage, we propose a novel approach to form a binary descriptor. Building upon the descriptor of ORB SLAM 2, we introduce a Gaussian blurring step followed by a comparison of the average gray size of pixel blocks in BRIEF pattern point pairs to generate the binary descriptor. The pixel block difference between feature point pixel blocks and BRIEF pattern points is calculated to form a dynamic threshold, and then the gray difference between the pattern pixel block pairs is compared with the dynamic threshold to form a new binary code string. The two feature descriptors mentioned above are fused into a 64-bit binary descriptor, which makes the feature descriptors less sensitive to noise.

Our work addresses the existing gaps by introducing an adaptive thresholding method for feature point detection and a novel binary descriptor formation technique. Unlike traditional methods, our approach considers local pixel intensity variations, making it more robust under varying lighting conditions. Additionally, our feature descriptor is less sensitive to noise,

addressing a common limitation in existing binary descriptors. This work represents a significant step towards achieving real-time, accurate, and robust feature point detection and description, particularly in the context of SLAM systems.

## Detection and description of feature points in the ORB algorithm

The ORB algorithm achieves robust and efficient detection and description of image feature points through fast corner extraction, orientation assignment, and binary descriptor generation. In the feature point extraction stage, the ORB algorithm employs an improved version of the FAST algorithm known as oriented FAST (oFAST). The feature point description stage utilizes the BRIEF algorithm with rotational invariance, referred to as Rotation-Aware Brief (rBRIEF). One of the key strengths of the ORB algorithm is its ability to deliver fast processing speed while maintaining scale and rotational invariance. Additionally, the algorithm exhibits robustness against noise and image transformations to a certain extent, further enhancing its overall reliability.

### oFAST algorithm

FAST is an efficient feature point (corner point) detection algorithm that does not involve feature point description. Taking the candidate feature point $p$ as the central reference of the circle, the 16 pixels of the circle formed by the radius of 3 pixels around it are calculated and compared with the point $p$. When there are enough pixel points with grayscale values that differ from the center point $p$, the image point $p$ may be a corner point. The algorithm consists of the following steps:

1. Choose any point $p$ in the image and assume the intensity is $I_p$. A circle is drawn with the center of point $p$ and a radius of 3. In Fig 1, the circle consists of 16 pixels. A candidate corner point $p$ is identified if there are $N$ consecutive pixels (usually 12 pixels) on the circle template that are either brighter than a threshold value plus $t$ or less bright than $I_p - t$. To optimize efficiency in practical applications, only the pixel differences between the center $p$ and the four pixels at positions 1, 5, 9, and 13 (representing the four cardinal directions) are examined for corner detection. If at least three of their absolute values exceed the threshold value, they are further compared with other pixels; otherwise, they are directly discarded. All pixel points in the image are traversed using the above method to obtain the candidate feature points for initial screening

2. Non-maximum suppression of the image: To identify if there are multiple feature points in a neighborhood (e.g., 3x3 or 5x5) centered on feature point $p$, a comparison is made. In this

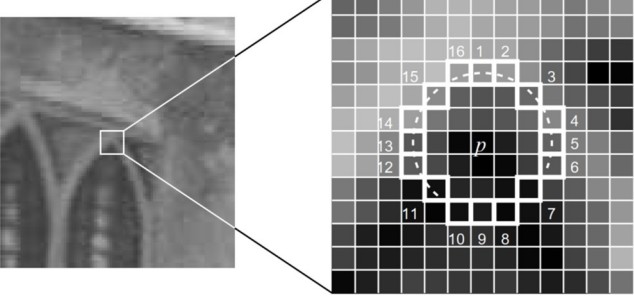

**Fig 1. FAST corner extraction diagram.**

comparison, the FAST score value (referred to as the *s*-value) is calculated separately for each feature point. The *s*-value is obtained by summing the absolute differences between the 16 points and the center. Subsequently, the feature point with the higher response value is preserved while suppressing the others. The score calculation formula is as follows (*V* is used in the formula to denote the score and *t* to denote the threshold value).

$$V = \max \{S(p) - S(v), S(v) - S(p)\}, \tag{1}$$

where: $S(p)$ is the cumulative accumulation of pixel intensities that are greater than or equal to $p$, and $S(v)$ represents the accumulation of pixel values that are less than $p$.

3. To achieve scale invariance, an image pyramid is constructed. Feature points are extracted at each level of the pyramid. These feature points from different levels are then mapped back to the original image, resulting in a collection of oFast feature points for the entire image. This approach ensures that the selected feature points maintain scale invariance.

4. The intensity centroid method is used to determine the feature point orientation and ensure the image rotational invariance. The specific calculation method is as follows: To ensure image rotational invariance and determine the orientation of feature points, the intensity centroid method [39] is employed. The calculation method for this purpose is as follows:

$$m_{pq} = \sum_{x,y \in B} x^p y^q I(x, y), \quad p, q = \{0, 1\}, \tag{2}$$

The centroid of an image block can be determined using moments.

$$C = \left( \frac{m_{10}}{m_{00}}, \frac{m_{01}}{m_{00}} \right), \tag{3}$$

The direction vector $\overrightarrow{OC}$, which is obtained by connecting the geometric center $O$ of the image block with the centroid $C$, represents the orientation of the feature point.

$$\theta = acr \tan \left( \frac{m_{01}}{m_{10}} \right), \tag{4}$$

To ensure rotational invariance, a common practice is to use an image block size of 31 31 pixels.

## rBRIEF feature descriptor

After acquiring feature points with orientation, BRIEF is employed to calculate the binary descriptors for these points. Within an image block, a localized self-centered patch serves as the space for $x$ and $y$ point pairs. A binary test $\tau$ is defined using the following equation.

$$\tau(p; x, y) := \begin{cases} 1 & : p(x) < p(y) \\ 0 & : p(x) \ge p(y) \end{cases}, \tag{5}$$

where $p(x)$ represents the intensity of $p$ at a point $x$.

By comparing the intensities of the two points, the features are encoded as $n$-bit binary vectors according to the binary descriptor:

$$f_n(p) := \sum_{1 \le i \le n} 2^{i-1} \tau(p; x_i, y_i), \tag{6}$$

The ORB algorithm improved the BRIEF descriptor. First, using the integral image in ORB, random point pairs are selected in a 31×31 pixels patch, and the selected random points are used as the center to calculate the gray mean value in a 5×5 sub-window, and the neighboring gray mean values of the random point pairs are compared for binary encoding, instead of deciding the encoding result only by the grays of two random point pairs, which can effectively solve the noise-sensitivity problem. Meanwhile, the BRIEF algorithm does not have rotational invariance, thus ORB adds a rotation factor called rBRIEF based on the original.

For an *n*-bit test binary feature description, define a $2 \times n$ matrix:

$$S = \begin{pmatrix} x_1, ..., x_n \\ y_1, ...y_n \end{pmatrix}, \tag{7}$$

The corresponding rotation matrix $R_\theta$ of angle $\theta$ is obtained by Eq (8). Subsequently, the matrix $S$ is rotated by $R_\theta$.

$$S = \begin{pmatrix} \cos\theta & \sin\theta \\ -\sin\theta & \cos\theta \end{pmatrix} S = R_\theta S, \tag{8}$$

In this case, a binary test descriptor with direction is obtained, which is defined as follows.

$$g_n(p, \theta) = f_n(p)|(x, y) \in Q_\theta, \tag{9}$$

In the above algorithm, the intensity of 256 pairs of pixel points is generally compared to generate a 32-byte feature descriptor.

## Related improvements

To enhance the ORB algorithm, our improvements primarily focus on the feature extraction and feature description phases. Firstly, we propose an adaptive threshold based on local image contrast. This improvement improves the algorithm's ability to detect distinctive points in areas with weak texture. Secondly, we employ an improved quadtree approach to optimize and manage the extracted feature points. In this study, different quadtree depths are set for images of varying scales based on the total number of feature points obtained from the pyramidal layers. The algorithm combines the grayscale difference information of the feature points themselves with randomly selected point pairs to generate binary descriptors. This inclusion provides additional spatial support information and enhances the discriminative power of the descriptors.

### Adaptive threshold determination

The traditional FAST algorithm employs a fixed threshold for feature point extraction, with more feature points extracted as the threshold decreases and vice versa. However, in the ORB algorithm, a predefined global threshold is artificially set for FAST, resulting in the extraction of numerous redundant feature points in regions with clear image changes. While a fixed threshold simplifies calculations and reduces processing time, it falls short in meeting the requirements of feature point extraction in various scenes. In scenarios where the image quality is poor or significant changes occur in the external environment, the number of extracted feature points can sharply decrease. This reduction can lead to map distortion or image stitching failure. To address this issue, Mur-Artal [32, 33] introduced a fixed dual-threshold method for feature extraction. When the high threshold fails to extract feature points, the low threshold is used for feature extraction. This dual-threshold mechanism effectively solves the problem of

insufficient feature point extraction under challenging conditions, ensuring a more robust and reliable feature detection process. However, the fixed artificially set threshold value still has some limitations and cannot be well adapted to different scenes.

The algorithm for the proposed adaptive threshold method is as follows. Similar to the grid-based approach used in the ORB SLAM2 system, we divide the image into grids and calculate the threshold for each grid using the following method.

$$Th = \alpha \sqrt{\frac{\sum_{1 \leqslant i \leqslant n}(I_{i_{\max}} - I_{i_{\min}})^2}{n}}, \tag{10}$$

where $I_{i_{\max}}$ and $I_{i_{\min}}$ are the $i$-th largest gray value and the $i$-th smallest gray value within each grid of the image, respectively; $n$ is the first $n$ largest (smallest) gray values in the image, and $n$ is taken as 10.$\alpha$ is the scale factor, generally taken as 0.15 to 0.30, which represents 15% to 30% of the absolute contrast of the image.

## Feature point homogenization strategy

The original ORB algorithm tends to focus on regions with rich texture, resulting in feature point aggregation and redundancy in areas with less or no texture. To address this issue and achieve a more uniform distribution of feature points throughout the image, this article employs the following homogenization methods:

- Step 1. In order to obtain multi-resolution scale information, we convert the input RGB image to grayscale, build the image pyramid, and use the Adaptive Threshold Method proposed earlier in this article to extract feature points on each layer. The number of feature points for each layer is:

$$N_\alpha = N \frac{\{1 - [s^2]\}}{\{1 - [s^2]^n\}} [s^2]^\alpha, \tag{11}$$

  In the equation, $N$, $n$, $N_\alpha$, and $s$ represent the expected total feature points, the pyramid layers, the feature points to extract, and the inverse scaling factor of each pyramid image, respectively. We use empirical values of 8 and 1.2 for $n$ and $s$.

- Step 2. Use the whole image as the initial extraction node for quadtree segmentation. Generally, for a 640*480 pixel image, the node count is initially 1.

- Step 3. Identifying all nodes in the image. If a node has no feature points, it is deleted. If it has only one feature point, the node is considered as a terminal node and is not further divided. However, if a node has more than one feature point, it continues to split into four children nodes.

- Step 4. Continue executing Step 3 iteratively until the number of nodes in the image reaches the specified number of feature points $N_\alpha$, and terminate the splitting process at that point.

- Step 5. Select the most salient feature point in each node using the Harris response value. If a node has more than one feature point, pick the one with the highest value as the node's representative point.

However, the above steps do not limit the quadtree split depth, which increases the algorithm's computation time. To reduce redundant feature computation, we set different depths based on the total feature points extracted from each pyramid layer. Algorithm 1 shows the detailed process. In Algorithm 1, $I_{ter}$ is the current quadtree's iteration number or split depth; $D_{max}$ is the maximum depth for the current pyramid layer; $N_{kp}$ is the node feature point count;

*bFinish* is the flag to stop splitting; and $N_{store}$ is the node storage count. The quadtree's maximum depth should follow this equation:

$$I_{ini}4^{I_{ter}} \geqslant N_{\alpha}, \tag{12}$$

$I_{ini}$ indicates the number of initial extraction nodes. Depending on the aspect ratio of the input image, $I_{ini}$ is usually 1 or 2. For example, the image in the KITTI dataset is 1241*376 pixels, so the initial number of nodes is 2.

**Algorithm 1:** Improved quadtree method algorithm

```
Data: Input: Image
Result: Output: ORB Keypoints Extracted
1 Construct the image pyramid;
2 for each pyramid layer i do
3    Adaptive threshold-based ORB feature point extraction;
4    Split quadtree nodes on level i pyramid;
5    for each subnode j do
6      while !bFinish do
7        if N_kp(j) > 1 and I_ter < D_max then
8           Divide child quadtree nodes;
9        end
10        if N_kp = 1 then
11           Store child node;
12        end
13      end
14      if N_store > N_α then
15        bFinish = true;
16      end
17    end
18    Retain features with large Harris response value;
19 end
```

## Feature description method combined with gray difference information

Our work references patch-based approaches in recognition. The texture segmentation system in [40] demonstrated the ability to detect local texture features by analyzing the cross-correlation of a central patch with nearby patches. In [41], a descriptor was formed by comparing a central patch with surrounding patches, thereby extending the shape-context [42] descriptor for intensity images. This descriptor has shown strong invariance to image style and local appearance.

In [43], two related descriptor families were proposed, namely the Three-Patch LBP descriptor and the Four-Patch LBP descriptor. LBP, initially introduced as a global representation for the entire image by [43–45], has proven to be a successful method for describing texture features. One specific application is the generation of TPLBP (Three-Patch Local Binary Pattern) codes, where the values of three patches are compared to generate a single bit value in the assigned code for each pixel. Similarly, FPLBP (Four-Patch Local Binary Pattern) codes are generated by comparing the values of four patches to produce a single-bit value in the assigned code for each pixel. While the algorithm proposed in [43] is primarily designed for face recognition and may not be directly applicable to image matching, its core concept involves generating feature descriptions by comparing pixel patches. Furthermore, in [36], the ORB-TPLGD algorithm utilizing the Three-Patch Method and local gray difference was introduced. Chaoqun Ma et al. [46] proposed a Quadtree ORB (QTORB) algorithm to enhance the feature point extraction ability in uniform regions. This paper refers to the feature description method that combines gray difference features from [46], and redesigns the algorithm by incorporating the

gray information of the feature points themselves. Hence, in this paper, we used a feature descriptor creation method that utilizes both grayscale intensity values and pixel patch difference information.

In the feature description phase, the ORB algorithm's descriptors primarily focus on the size relationships of random pixel dot pairs within the local area of the feature point. This means that the distribution of size relationships among the selected pixel dot pairs is crucial. However, the algorithm overlooks the actual gray values of the pixels themselves and only takes into account the size relationships of the random dot pairs.

In this paper, based on the improved BRIEF in ORB, we propose to use the size and the difference of gray values for feature point description. We introduce the intensity difference information to construct the descriptors and focus on the gray information around the feature points. The specific schematic diagram is shown in Fig 2. Assuming there is a candidate feature point $P$ on the smoothed image, a specific pattern is used to select $n$ pairs of pixel blocks for grayscale comparison, resulting in the generation of binary codes, denoted as $f_n(p)$.

Additionally, to generate another binary code, the average difference between each pixel block and the feature point pixel block (Fig 2) is calculated. The gray difference $H_i$ can be defined as:

$$\{H_i\}_{i=1...n} = \{|g(\bar{A}_i) - g(\bar{B}_i)|\}_{i=1...n}, \tag{13}$$

where $g$ denotes the grayscale sum of the pixel block and $\bar{g}$ denotes the average gray level of the pixel block. Each feature point has $n$ corresponding gray differences. Binary code conversion requires non-floating point data. Therefore, to generate a binary code, it is necessary to determine a threshold value and compare the gray difference value with this threshold. By calculating the gray differences and evaluating them against the threshold, the binary code for each

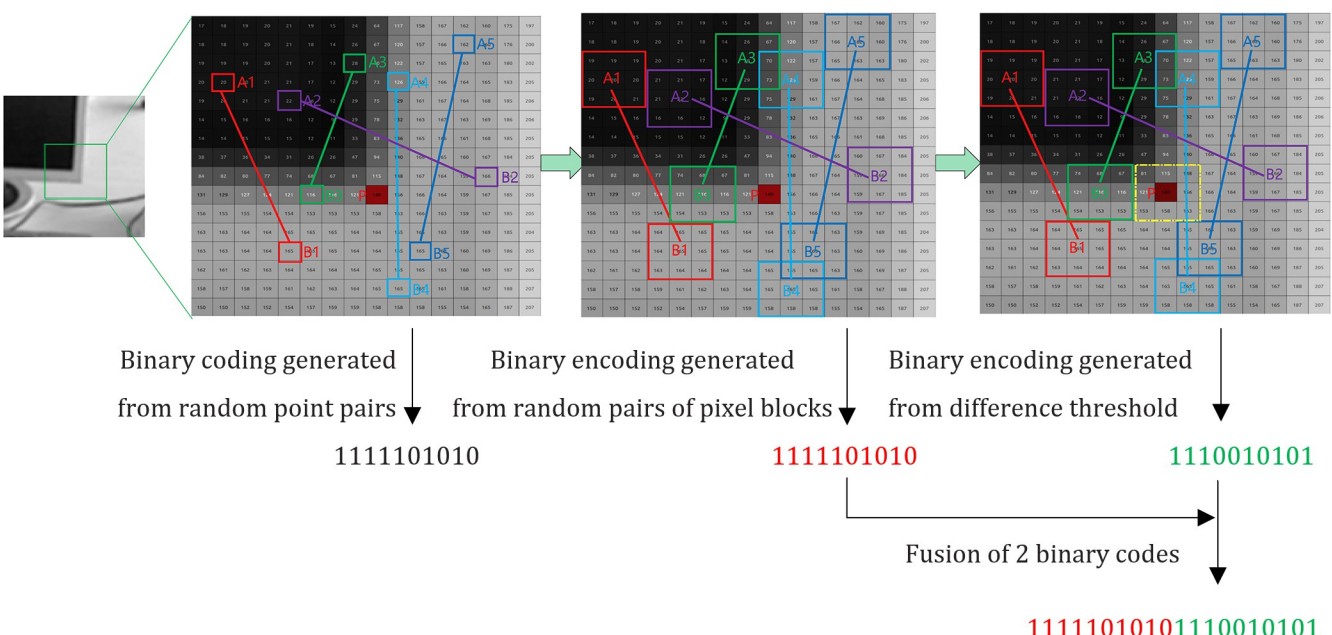

**Fig 2. Binary descriptor of the combination of gray size and gray value difference information.**

feature point can be derived. The threshold value $T_{average}$ can be defined as:

$$T_{average} = \frac{\{|g(\bar{A}_i) - g(\bar{P})| + |g(\bar{B}_i) - g(\bar{P})|\}_{i=1\dots n}}{2n}, \tag{14}$$

Compare $n$ gray difference and threshold $T_{average}$:

$$\tau\left(H_i, T_{average}\right) := \begin{cases} 1, & H_i > T_{average} \\ 0, & otherwise \end{cases}, \tag{15}$$

According to Eq (15), $f_n(p)^{\wedge}$ is generated:

$$f_n(p)^{\wedge} := \sum_{1 \leqslant i \leqslant n} 2^{i-1}\tau\left(H_i, T_{average}\right), \tag{16}$$

In the procedure depicted in Fig 2, the two binary descriptors, $f_n(p)$ and $f_n(p)^{\wedge}$, are merged together to form a novel binary descriptor.

$$Description\,(p) = f_n\,(p) + f_n\,(p)^{\wedge}, \tag{17}$$

## The overall workflow of our algorithm

In this paper, the ALGD-ORB algorithm enhances three key stages of ORB: feature extraction, feature point allocation, and feature point description. The complete procedure of the algorithm is outlined in Fig 3, while Algorithm 2 provides a breakdown of the specific steps involved in ALGD-ORB.

**Algorithm 2:** Detailed explanation of ALGD-ORB operation

```
Data: Input: Image
Result: Output1: Keypoints, Output2: Keypoints description
1 Pre-compute the scale pyramid based on the input gray image. The
total number of pyramid layers and the scaling factor are set to
empirical values of 8 and 1.2
2 Gridding each layer of the pyramid image. Set the size of the image
grid to, calculate the adaptive threshold Th according to Eq 10, tra-
verse and calculate the feature points in all grids of all pyramid
images, and note that the extracted N feature points are p = {p₁,
p₂,..., pᵢ, ..., p_N};
3 Use the improved quadtree method to average and distribute feature
points in each image pyramid layer;
4 The number of all feature points after homogenization by the
improved quadtree algorithm is denoted as K. Then P = {P1, P2,..., PK};
5 for i = 1, 2, ..., K do
6   Define a 31 × 31 pixel image block M centered on Pᵢ;
7   Select random point pairs in, and the coordinates of each random
point relative to the feature point are R = {Rₜ}ₜ₌₁…ₙ = {[(xₜ₁, yₜ₁),
(xₜ₂, yₜ₂)]}ₜ₌₁…ₙ;
8   Define 5 × 5 pixel patches Aₜ₁, Bₜ₁, Pᵢ centered on point (xₜ₁, yₜ₁),
(xₜ₂, yₜ₂) and pᵢ;
9   for t = 1, 2, ..., n do
10     if g(Aₜ₁) < g(Bₜ₂) then
11       Assign 1 to corresponding bit;
12     else
13       Assign 1;
14     end
15   end
16   Calculate the gray difference Hᵢ and threshold T_average;
```

```
17  for t = 1, 2, ..., n do
18    if H_i > T_average then
19      Assign 1 to corresponding bit;
20    else
21      Assign 1;
22    end
23  end
24    The final Description(p) = f_n (p) + f_n (p)^ for each feature point
is obtained by joining f_n (p) and f_n (p)^;
25 end
```

## Experiments and results

All experiments were based on a processor model i7-11800H @ 2.30GHz, 16GB memory size, and Ubuntu20.04 operating system. The simulation environment is Microsoft Visual Studio code 2021 and the simulation language is C++ combined with OpenCV4.2.0. The dataset tested is the Oxford dataset [46].

In this study, we compare the ALGD-ORB algorithm with several existing methods, including SIFT, SURF, BRISK, FREAK, LATCH, LDB, ORB, LF-Net, D2-Net, R2D2, and the algorithm proposed in ORB SLAM2 (referred to as Mur-ORB). For feature point detection, SIFT, SURF, BRISK, and FREAK employ their respective detection techniques, while LATCH, LDB, ORB, Mur-ORB, and ALGD-ORB utilize the FAST corner point detection method. Regarding feature point matching, SIFT, SURF, LF-Net, D2-Net, and R2D2 use Euclidean distance as their matching metric since their feature vectors are represented by floating-point values. On the other hand, BRISK, FREAK, LATCH, LDB, ORB, Mur-ORB, and ALGD-ORB utilize Hamming distance to measure similarity. More detailed information on each algorithm can be found in Table 1. Additionally, the RANSAC algorithm is employed to evaluate all algorithms, considering a pair of key points as a correct match if the matching error falls within 3 pixels.

### Evaluation metrics

In the experiments, four evaluation criteria were utilized to assess the algorithm's performance. The first criterion is Evenness [47], which involves dividing the image into 10 regions: top, bottom, left, right, top left, bottom right, top right, bottom left, center, and periphery. These regions are derived from four directions (vertical, horizontal, 45˚, and 135˚) as well as the

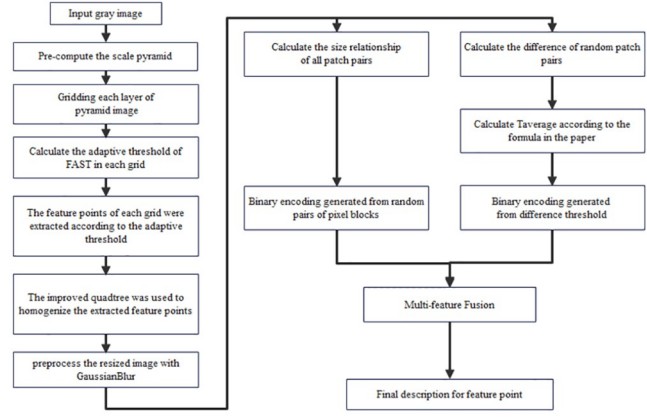

**Fig 3. Illustrates the comprehensive workflow of the ALGD-ORB algorithm.**

**Table 1. Details of each algorithm.**

| Tested algorithms | Detection method | Description method | Matching distance | Description size | Error threshold | Mismatch elimination |
|---|---|---|---|---|---|---|
| SIFT | DOG | SIFT | Euclidean | 128-dimension | 3 pixels | RANSAC |
| SURF | Hessian | SURF | Euclidean | 64-dimension | 3 pixels | RANSAC |
| BRISK | BRISK | BRISK | Hamming | 64-byte | 3 pixels | RANSAC |
| FREAK | Harris | FREAK | Hamming | 64-byte | 3 pixels | RANSAC |
| LATCH | FAST | LATCH | Hamming | 32-byte | 3 pixels | RANSAC |
| LDB | FAST | LDB | Hamming | 64-byte | 3 pixels | RANSAC |
| ORB | FAST | BRIEF | Hamming | 64-byte | 3 pixels | RANSAC |
| Mur-ORB | FAST | BRIEF | Hamming | 32-byte | 3 pixels | RANSAC |
| LF-Net | LF-Net | LF-Net | Euclidean | 128-dimension | 3 pixels | RANSAC |
| D2-Net | D2-Net | D2-Net | Euclidean | 128-dimension | 3 pixels | RANSAC |
| R2D2 | R2D2 | R2D2 | Euclidean | 128-dimension | 3 pixels | RANSAC |
| ALGD-ORB | FAST | ALGD-ORB | Hamming | 64-byte | 3 pixels | RANSAC |

center and periphery (as depicted in Fig 4). The number of feature points within each region is counted, and the variance (V) of the group is calculated based on these data. Subsequently, the uniformity is determined using a specific formula.

$$u = 101 \times log(V), \tag{18}$$

A smaller value indicates a more balanced distribution of feature points across different regions, indicating better uniformity in their distribution.

The second evaluation metric is precision [48], which calculates the ratio of the number of correct matches $Num_{correctmatches}$ to the total number of matched points $Num_{allmatches}$. The equation for precision is as follows:

$$Precision = \frac{Num_{correctmatches}}{Num_{allmatches}} \times 100 \tag{19}$$

The root mean square error (RMSE) is a third evaluation metric that measures the overall positional accuracy between detected and true matching points in image matching. By calculating the differences between matched point pairs, RMSE quantifies the error between the matching results and the ground truth. A lower RMSE value indicates a higher accuracy and stability of the image matching algorithm. Consider a pair of matching points $A_a(x_a, y_a)$ and $B_b(x_b, y_b)$ obtained by an image matching algorithm. Here, $A_a(x_a, y_a)$ represents a feature point in image A, and $B_b(x_b, y_b)$ represents a feature point in image B. The true position of feature point $A_a(x_a, y_a)$ in image B is denoted as $\hat{B}_b(\hat{x}_b, \hat{y}_b)$. The RMSE calculation formula for this

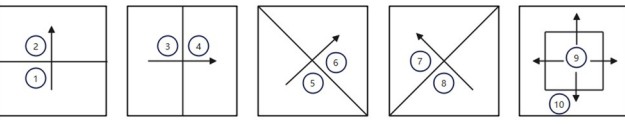

**Fig 4. Schematic diagram of five-directional segmentation.**

pair of points is as follows:

$$RMSE = \sqrt{(x_b - \hat{x}_b)^2 + (y_b - \hat{y}_b)^2}, \tag{20}$$

The fourth evaluation metric focuses on the computational efficiency of the image matching algorithm. It includes various processing times, such as feature point extraction, descriptor generation, and matching. Notably, tasks like image loading and the calculation of other algorithm performance metrics are excluded from this evaluation.

## Oxford optical image dataset

To validate the proposed algorithm's effectiveness, we utilized images from a publicly available database established by K. Mikolajczyk and C. Schmid [48]. This comprehensive database comprises eight distinct image categories, encompassing variations such as blur (Trees and Bikes), lighting (Leuven), JPG compression (Ubc), scale and rotation (Bark and Boat), and visual enhancements (Wall and Graf). Refer to Fig 5 for a visual representation of these categories. Within each category, the first image can be paired with any of the other five images to form input image pairs. The dataset provides the homography matrix (referred to as H1to2p) between the two images in each pair, enabling the calculation of ground truth information for comparison purposes.

## Experiment result based on Oxford dataset

The experiment was conducted on eight image categories from the dataset, including Bikes (blur), Trees (blur), Graf (viewpoint), Wall (viewpoint), Bark (zoom and rotation), Boat (zoom and rotation), Leuven (light), and Ubc (JPEG compression). For each image pair, 30 experiments were conducted using the feature extraction algorithms proposed in Table 1. To ensure generality, the average value of the 30 experiments was used as the experimental result.

**Impact of feature points distribution on matching performance: A study on the Oxford dataset.** In this study, feature point detection and matching algorithms including SIFT, SURF, BRISK, FREAK, LATCH, LDB, ORB, Mur-ORB, LF-Net, D2-Net, R2D2 and ALGD-ORB were selected to conduct experiments on feature point extraction and image matching using images from the Oxford dataset. In addition to evaluating the performance differences of various feature point detection and matching algorithms on the Oxford dataset, we also analyzed the distribution of feature points using each algorithm to intuitively observe the performance differences in feature point uniformity distribution.

Fig 6 provides a visual representation of the experimental results obtained by testing nine different algorithms on the Leuven image set from the Oxford dataset. It offers a scientific and experimental perspective on the results. The experimental results indicate that the performance of LATCH, LDB, ORB LF-Net and D2-Net algorithms differ from other algorithms in terms of feature point extraction. They tend to extract image regions with more prominent edges, while not extracting many feature points in homogenous regions. Although SIFT, SURF, BRISK, FREAK and R2D2 algorithms have the ability to extract feature points from homogenous regions, the density of feature points in such areas tends to be lower. While most feature points still tend to be concentrated in edge areas, resulting in significant overlap, the distribution of feature points in the Mur-ORB, ALGD-ORB, and R2D2 algorithms appears to be more uniform compared to other tested algorithms. Visual observation alone does not provide a clear distinction regarding which algorithm exhibits the most uniform distribution. However, according to Table 2, the ALGD-ORB algorithm demonstrates a comparatively more uniform distribution of feature points when compared to the Mur-ORB and R2D2

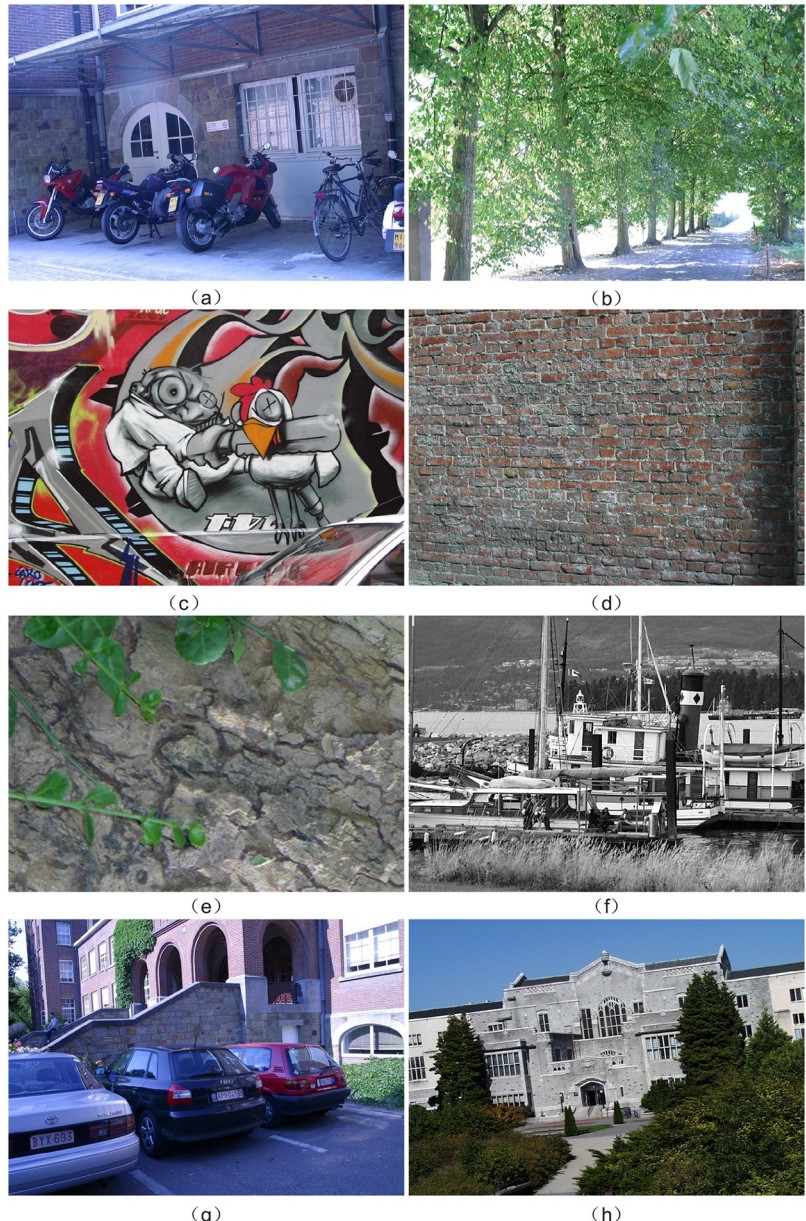

**Fig 5. Optical images from Oxford dataset.** (a) Bikes (Blur) (b) Trees (Blur) (c) Graf (Viewpoint) (d) Wall (Viewpoint) (e) Bark (Zoom & Rotation) (f) Boat (Zoom & Rotation) (g) Leuven (Light) (h) UBC (JPEG Compression).

algorithms. ALGD-ORB shows the ability to extract feature points from specific uniform regions where other algorithms may struggle.

Fig 7 presents the image matching results obtained using the nine tested algorithms on the Leuven image set from the Oxford dataset. The matching pairs are depicted with red lines for incorrect matches and green lines for correct matches after applying RANSAC. Similar to the feature point distribution observed in Fig 6, the feature point matches obtained from SIFT, SURF, BRISK, FREAK, LATCH, LDB, and ORB algorithms in Fig 7 exhibit a noticeable concentration. Conversely, the Mur-ORB, ALGD-ORB, LF-Net, D2-Net, and R2D2 algorithms

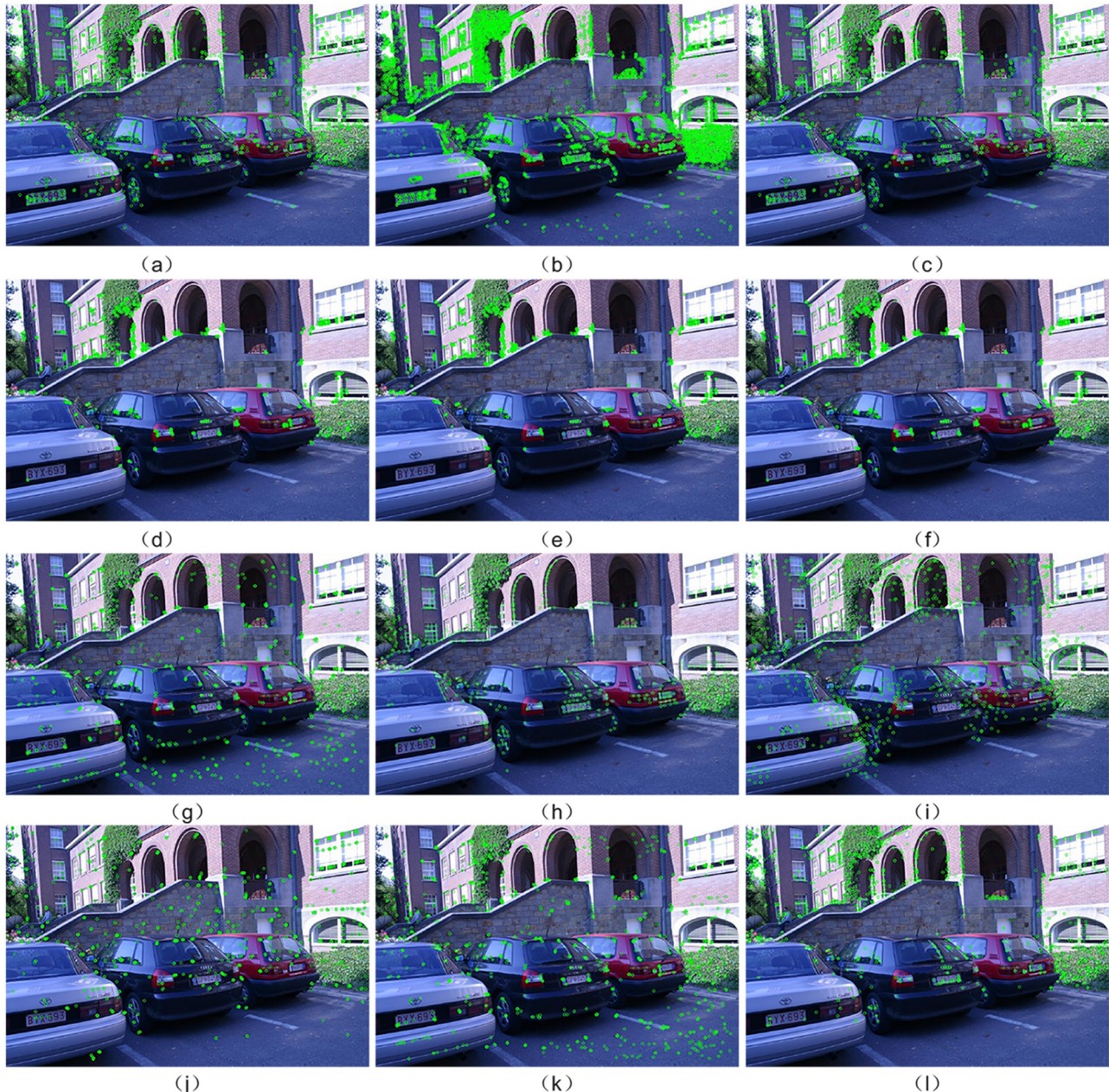

**Fig 6. Illustrates the distribution of feature points extracted by different algorithms.** (a) SIFT, (b) SURF, (c) BRISK, (d) FREAK, (e) LATCH, (f) LDB, (g) ORB, (h) Mur-ORB, (i) LF-Net, (j) D2-Net, (k) R2D2, and (l) ALGD-ORB, as demonstrated on the Leuven group image from the Oxford dataset.

produce more uniformly distributed matching results, accurately capturing matches in both uniform and edge regions. Notably, the ALGD-ORB algorithm demonstrates a higher degree of uniformity in the matching results compared to the Mur-ORB and R2D2 algorithms. For a more accurate analysis of algorithm performance, statistical analysis is conducted on the feature point distribution uniformity across all images in the Oxford dataset, as presented in Table 2.

**Table 2. Average feature point distribution uniformity across image groups in the Oxford dataset.**

| Method | Bark | Bikes | Boat | Graf | Leuven | Trees | Ubc | Wall |
|---|---|---|---|---|---|---|---|---|
| SIFT | 427.7493 | 374.9600 | 448.1905 | 415.6467 | 460.4822 | 444.5727 | 430.3688 | 415.4240 |
| SURF | 384.6193 | 378.0553 | 492.6830 | 412.0660 | 435.0193 | 501.0945 | 467.9445 | 467.9645 |
| BRISK | 521.5460 | 411.5352 | 637.9578 | 537.0008 | 521.7793 | 597.7933 | 597.3543 | 622.9485 |
| FREAK | 402.0787 | 372.1875 | 500.9532 | 451.3865 | 375.7668 | 506.9070 | 484.8492 | 486.5585 |
| LATCH | 470.0743 | 438.8417 | 463.3097 | 481.6723 | 478.7010 | 467.3087 | 455.6300 | 430.0753 |
| LDB | 469.7415 | 443.3937 | 463.9257 | 478.5605 | 478.4625 | 463.1293 | 456.6222 | 433.1270 |
| ORB | 470.0743 | 438.8417 | 463.3097 | 481.6723 | 478.7010 | 467.3087 | 455.6300 | 430.0753 |
| MUR-ORB | **363.3928** | 326.1843 | 371.9970 | 332.3045 | **289.8742** | 389.6325 | 383.6460 | 395.2780 |
| LF-Net | 409.2150 | 400.3467 | 387.7283 | 325.7100 | 434.4850 | 422.8600 | 390.0633 | 414.0567 |
| D2-Net | 471.1390 | 412.8557 | 475.4643 | 375.9733 | 495.4330 | 430.8253 | 455.2507 | 495.7543 |
| R2D2 | 466.9467 | 457.0817 | 429.9083 | 401.8250 | 410.1850 | 434.1800 | 432.4667 | 443.5567 |
| ALGD-ORB | 390.5125 | **266.9185** | **368.9225** | **311.4857** | 330.4555 | **367.1153** | **368.3155** | **377.0120** |

Table 2 demonstrates the performance of different algorithms across the eight scene transformations available in the Oxford dataset. Both Mur-ORB and ALGD-ORB exhibit superior uniformity in feature point extraction compared to the other algorithms tested. While ALGD-ORB may show slightly lower uniformity than Mur-ORB in specific scenarios such as the Bark and Leuven group images, it consistently achieves more uniform feature point distributions across the remaining six image groups. Notably, ALGD-ORB demonstrates the ability to extract feature points in certain uniform regions where Mur-ORB falls short. Compared to Mur-ORB, ALGD-ORB demonstrates better adaptability under multiple scenes.

**Matching precision analysis of the Oxford dataset.** This study conducted image matching tests on all images within the Oxford dataset. Each image group consisted of five pairs of images that required matching (ranging from 1-2 to 1-6, with increasing variations between the pairs). Fig 8 presents the matching accuracy for each image pair, while the symbol "Mean" represents the average accuracy of image matching across the eight image groups. Further details regarding the average accuracy can be found in Table 3.

Based on Fig 8 and Table 3, we can see that for the Wall image set, SIFT achieves the highest precision. For the Bark image set, BRISK achieves the highest precision. For the Bikes, Graf, and Leuven image sets, the R2D2 algorithm has the highest precision. For the Boat image set, ORB achieves the highest precision. For the Trees and Ubc image sets, Mur-ORB has the highest precision. Mur-ORB also achieves the highest average precision.

The ALGD-ORB algorithm is an improved and optimized version based on the ORB algorithm. However, it does not demonstrate exceptional performance in certain image sets. The main reason for this is that, in order to improve computational efficiency, the feature detector focuses only on small image regions and specifically on low-level structures such as corner points. Subsequently, the descriptors capture higher-level information in the larger surrounding regions of the keypoints. In the presence of strong appearance variations, the detection of keypoints becomes unstable due to the susceptibility of the low-level information used by the detector to variations in low-level image statistics, such as pixel intensity. Nevertheless, despite the lack of significant advantages in certain image sets, the ALGD-ORB algorithm still exhibits acceptable performance in terms of accuracy.

**RMSE analysis of image matching on the Oxford dataset.** According to Fig 9, the RMSE (Root Mean Square Error) metrics of SIFT, SURF, BRISK, FREAK, LATCH, LDB, ORB, Mur-ORB, LF-Net, D2-Net, R2D2, and ALGD-ORB algorithms on the Oxford dataset under

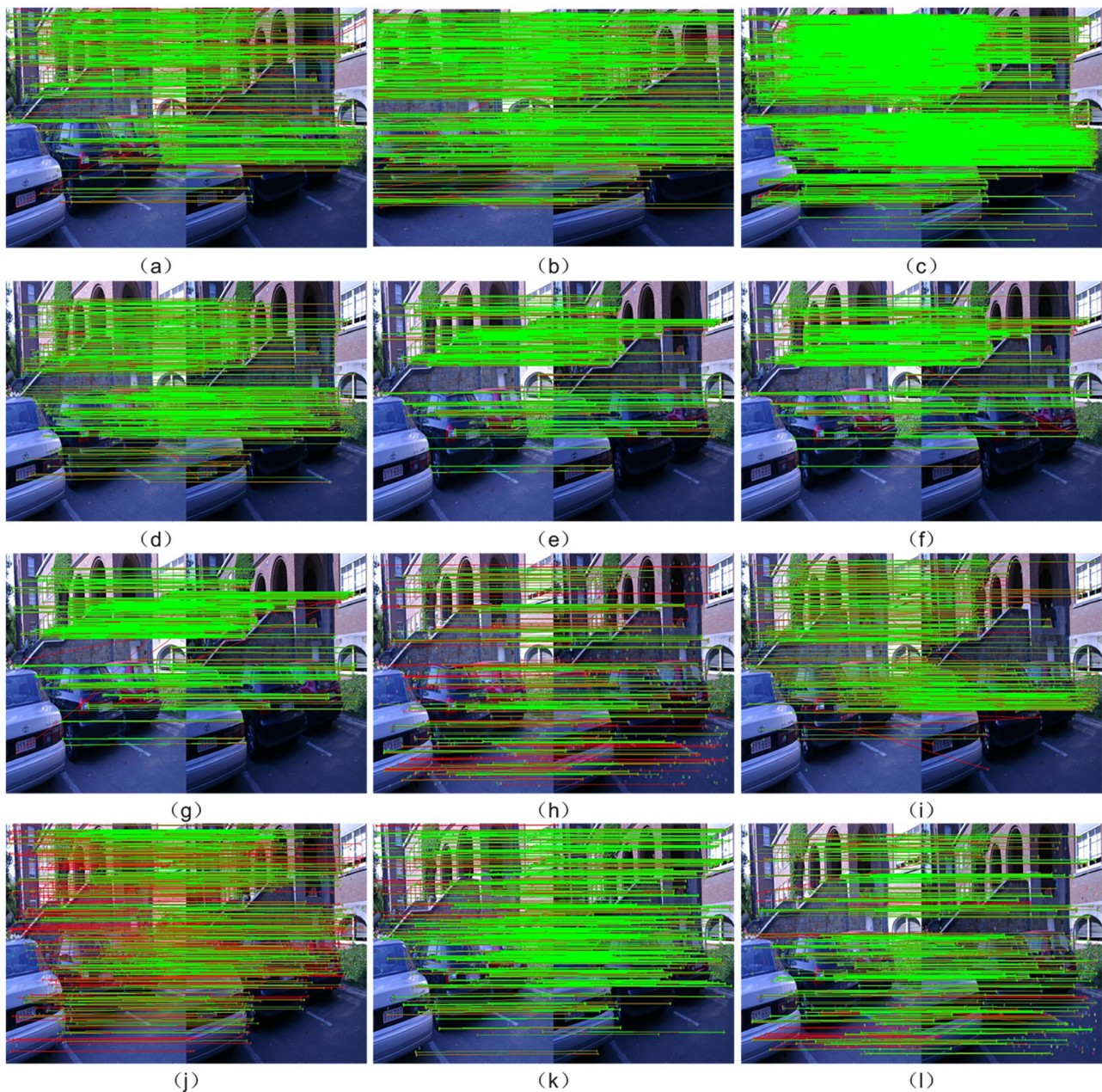

**Fig 7. Illustrates the distribution of feature points extracted by different algorithms.** (a) SIFT, (b) SURF, (c) BRISK, (d) FREAK, (e) LATCH, (f) LDB, (g) ORB, (h) Mur-ORB, (i) LF-Net, (j) D2-Net, (k) R2D2, and (l) ALGD-ORB, as demonstrated on the Leuven group image from the Oxford dataset.

different image variation scenarios are presented. In Table 4, LF-Net algorithm has the smallest average RMSE value in the Bikes image set. R2D2 algorithm achieves the smallest average RMSE value in the Boat image set. Similarly, in the Leuven and Ubc image sets, R2D2 algorithm exhibits the smallest average RMSE value. ALGD-ORB algorithm demonstrates the smallest average RMSE values in the Bark, Graf, Trees, and Wall image sets. However, the RMSE value for the Bark image set is missing for the R2D2 algorithm, indicating that it failed to match correct features in this particular image set, resulting in a considerably large RMSE.

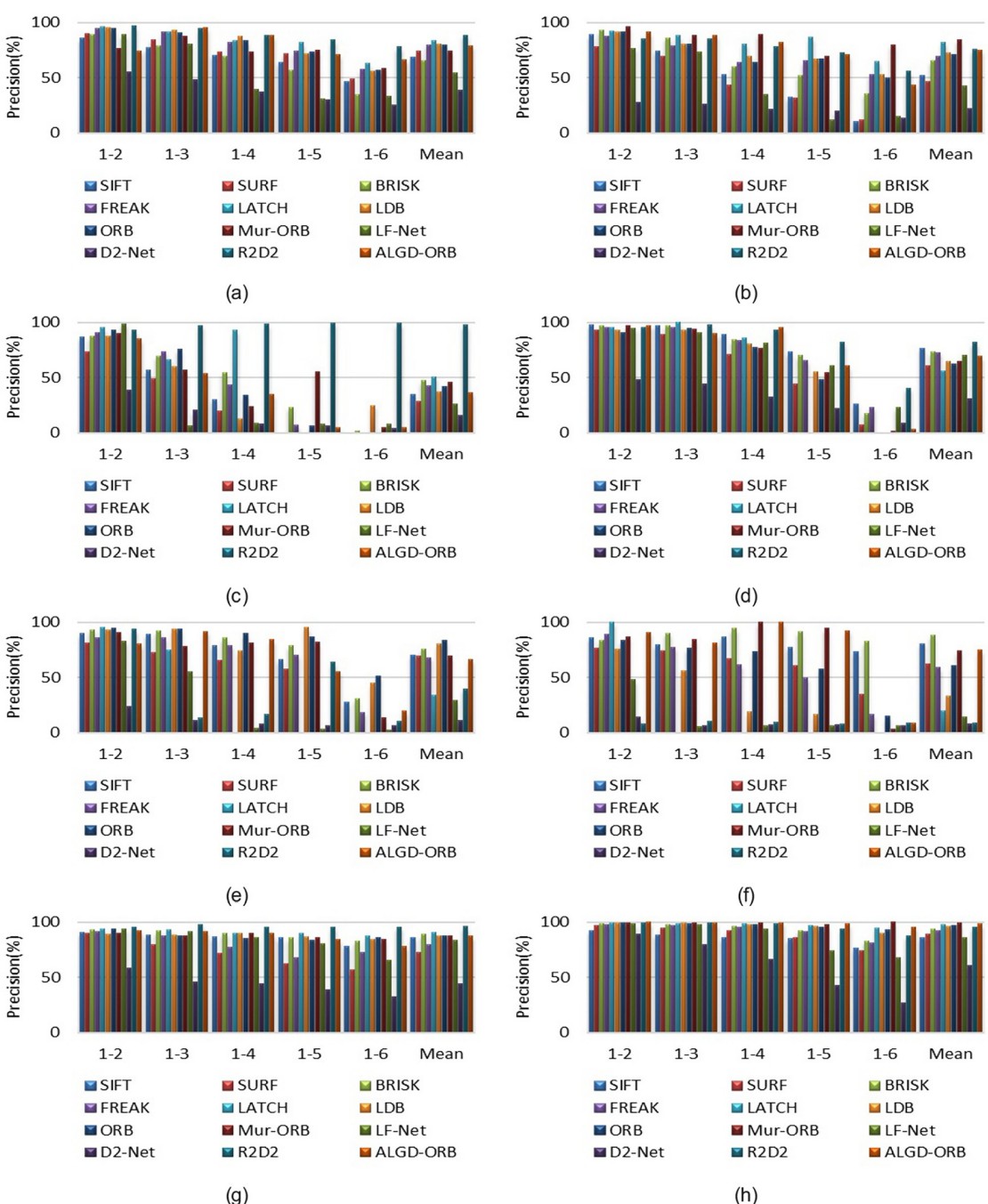

**Fig 8. Matching precision results on the Oxford dataset.** (a) Bikes (blur). (b) Trees (blur). (c) Graf (viewpoint). (d) Wall (viewpoint). (e) Bark (zoom and rotation). (f) Boat (zoom and rotation). (g) Leuven (light). (h) Ubc (JPEG compression).

Overall, the ALGD-ORB algorithm exhibits the smallest average RMSE value of 1.0285 across all image sets. This suggests that the ALGD-ORB algorithm performs well in terms of feature point extraction in various image variation scenarios within the Oxford dataset.

**Matching operation time analysis on the Oxford dataset.** To demonstrate the efficiency of the ALGD-ORB algorithm, we compared the average runtime of various algorithms,

**Table 3. Mean precision for image groups based on the Oxford dataset.**

| Method | Bark | Bikes | Boat | Graf | Leuven | Trees | Ubc | Wall | Mean |
|---|---|---|---|---|---|---|---|---|---|
| SIFT | 80.8725 | 68.9463 | 70.4639 | 34.8244 | 86.1646 | 52.0260 | 85.8436 | 76.8779 | 69.5024 |
| SURF | 62.7796 | 74.0554 | 57.2284 | 28.5542 | 72.3417 | 46.9106 | 88.9931 | 61.0717 | 61.4918 |
| BRISK | **88.4577** | 65.9645 | 76.2538 | 47.4627 | 88.9132 | 65.5764 | 93.8842 | 73.3165 | 74.9786 |
| FREAK | 58.9964 | 80.0443 | 68.1236 | 43.0892 | 79.5462 | 69.9329 | 92.7578 | 72.7909 | 70.6602 |
| LATCH | 20.0000 | 83.5060 | 34.1469 | 51.0395 | 90.9826 | 82.5729 | 97.6267 | 56.3794 | 64.7068 |
| LDB | 33.5714 | 80.8975 | 80.3537 | 37.0953 | 87.8256 | 72.4015 | 96.5186 | 64.6344 | 69.1623 |
| ORB | 61.2977 | 80.0802 | **83.4492** | 41.9211 | 87.5144 | 70.7893 | 97.1422 | 62.3779 | 73.0715 |
| Mur-ORB | 73.9811 | 74.4054 | 69.3961 | 46.2989 | 87.5545 | **84.8358** | **99.3537** | 64.8014 | **75.0784** |
| LF-Net | 14.4980 | 54.5980 | 29.7260 | 26.0360 | 83.5420 | 42.5240 | 86.5100 | 70.3500 | 50.9730 |
| D2-Net | 8.3920 | 39.1680 | 11.1280 | 15.7420 | 44.0100 | 21.9920 | 61.2160 | 31.3000 | 29.1185 |
| R2D2 | 9.2560 | **85.5180** | 39.6640 | **80.6640** | **95.2060** | 75.5220 | 95.9900 | **81.9420** | 70.4725 |
| ALGD-ORB | 74.7929 | 79.2037 | 66.2713 | 36.7969 | 87.3403 | 75.2477 | 98.6568 | 69.3676 | 73.4597 |

including SIFT, SURF, BRISK, FREAK, LATCH, LDB, ORB, Mur-ORB, LF-Net, D2-Net, R2D2, and ALGD-ORB, on the Oxford dataset. As shown in Fig 10 and Table 5, the ALG-D-ORB algorithm has a remarkable runtime efficiency. during runtime. Among the tested algorithms, the R2D2 algorithm, based on deep learning, has the longest runtime, while the ORB algorithm has the shortest. Deep learning-based feature extraction algorithms generally require more time compared to traditional feature detection algorithms. Despite the improvements made to the FREAK, LATCH, LDB, and Mur-ORB algorithms derived from the ORB algorithm, they still consume more time. In comparison to the Mur-ORB algorithm, the ALG-D-ORB algorithm experiences a slight increase in runtime due to the inclusion of additional processes, such as adaptive threshold selection and 64-bit descriptor generation.

Overall, the ALGD-ORB algorithm demonstrates excellent efficiency on the Oxford dataset. Despite the slight increase in runtime compared to the Mur-ORB algorithm, the ALGD-ORB algorithm maintains competitive performance while incorporating additional processes for adaptive threshold selection and 64-bit descriptor generation. The advantages of the

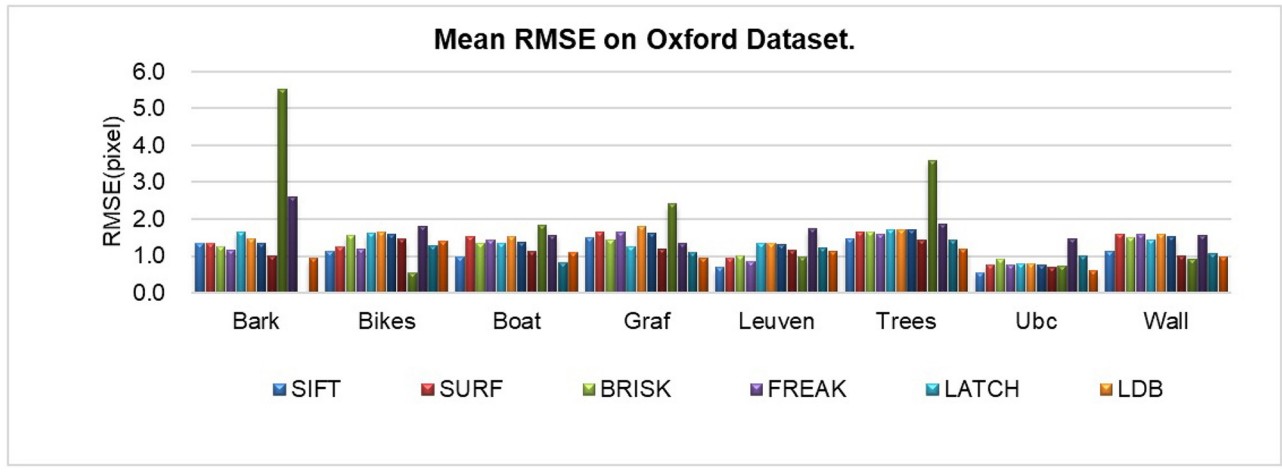

**Fig 9. Mean RMSE analysis on the Oxford dataset.**

**Table 4. Feature points are extracted from images with various scenes and lighting conditions.** The mean RMSE of each group is shown in units of pixels.

| Method | Bark | Bikes | Boat | Graf | Leuven | Trees | Ubc | Wall | Mean |
|---|---|---|---|---|---|---|---|---|---|
| SIFT | 1.3328 | 1.1102 | 0.9648 | 1.5059 | **0.6852** | 1.4728 | **0.5474** | 1.1307 | 1.0937 |
| SURF | 1.3266 | 1.2415 | 1.5226 | 1.6564 | 0.9243 | 1.6562 | 0.7657 | 1.5981 | 1.3364 |
| BRISK | 1.2320 | 1.5545 | 1.3392 | 1.4267 | 0.9915 | 1.6320 | 0.8958 | 1.4859 | 1.3197 |
| FREAK | 1.1584 | 1.1973 | 1.4222 | 1.6388 | 0.8417 | 1.5961 | 0.7630 | 1.5763 | 1.2742 |
| LATCH | 1.6482 | 1.6053 | 1.3382 | 1.2558 | 1.3523 | 1.7088 | 0.7733 | 1.4292 | 1.3889 |
| LDB | 1.4619 | 1.6352 | 1.5224 | 1.8016 | 1.3319 | 1.7189 | 0.7980 | 1.5917 | 1.4827 |
| ORB | 1.3524 | 1.5880 | 1.3734 | 1.6218 | 1.3086 | 1.7165 | 0.7529 | 1.5174 | 1.4039 |
| MUR-ORB | 1.0032 | 1.4523 | 1.1211 | 1.1992 | 1.1520 | 1.4186 | 0.6898 | 0.9953 | 1.1289 |
| LF-Net | 5.5360 | **0.5340** | 1.8240 | 2.4040 | 0.9600 | 3.5900 | 0.7320 | 0.9810 | 2.0625 |
| D2-Net | 2.5860 | 1.8020 | 1.5680 | 1.3500 | 1.7360 | 1.8680 | 1.4740 | 1.5560 | 1.7425 |
| R2D2 | - | 1.2680 | **0.8100** | 1.0880 | 1.2240 | 1.4180 | 1.0080 | 1.0760 | 1.1274 |
| ALGD-ORB | **0.9274** | 1.3909 | 1.0838 | **0.9519** | 1.1321 | **1.1733** | 0.5932 | **0.9752** | **1.0285** |

ALGD-ORB algorithm in terms of runtime efficiency are clearly depicted in Fig 10 and summarized in Table 5. In contrast to LF-Net, D2-Net, and R2D2, which are deep learning-based algorithms, the ALGD-ORB algorithm does not rely on GPU acceleration yet still provides efficient matching operations. The ALGD-ORB algorithm provides a balanced performance in terms of runtime and feature extraction capabilities, making it a suitable choice for a variety of applications, especially in re-source-constrained scenarios where GPU acceleration is not available.

## Discussion

To address the issue of dense and overlapping features extracted by the ORB algorithm, the Mur-ORB algorithm employs a quad-tree approach. This involves dividing the extracted

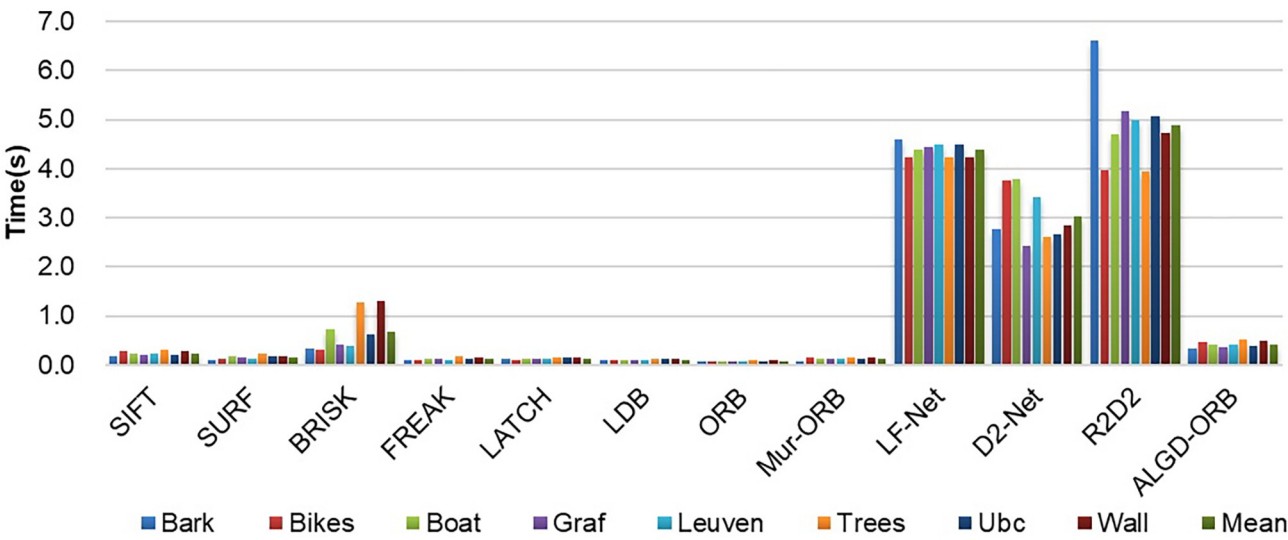

**Fig 10. Statistical analysis of mean operation time on the oxford dataset.**

**Table 5. Shows the mean operation time of each algorithm on Oxford dataset in units of seconds.**

| Method | Bark | Bikes | Boat | Graf | Leuven | Trees | Ubc | Wall | Mean |
|---|---|---|---|---|---|---|---|---|---|
| SIFT | 0.1783 | 0.2925 | 0.2449 | 0.2152 | 0.2241 | 0.3124 | 0.2166 | 0.2813 | 0.2457 |
| SURF | 0.1107 | 0.1211 | 0.1738 | 0.1545 | 0.1213 | 0.2306 | 0.1748 | 0.1881 | 0.1594 |
| BRISK | 0.3289 | 0.3188 | 0.7406 | 0.4096 | 0.3788 | 1.2866 | 0.6221 | 1.2984 | 0.6730 |
| FREAK | 0.1064 | 0.1090 | 0.1383 | 0.1213 | 0.1138 | 0.1803 | 0.1182 | 0.1652 | 0.1316 |
| LATCH | 0.1314 | 0.1164 | 0.1175 | 0.1207 | 0.1231 | 0.1476 | 0.1544 | 0.1435 | 0.1318 |
| LDB | 0.1060 | 0.1145 | 0.1074 | 0.1021 | 0.0940 | 0.1367 | 0.1174 | 0.1397 | 0.1147 |
| ORB | **0.0800** | **0.0889** | **0.0898** | **0.0842** | **0.0776** | **0.1044** | **0.0825** | **0.1014** | **0.0886** |
| Mur-ORB | 0.0904 | 0.1456 | 0.1361 | 0.1171 | 0.1202 | 0.1674 | 0.1208 | 0.1594 | 0.1321 |
| LF-Net | 4.6000 | 4.2200 | 4.3800 | 4.4500 | 4.4900 | 4.2400 | 4.5000 | 4.2400 | 4.3900 |
| D2-Net | 2.7600 | 3.7700 | 3.7800 | 2.4300 | 3.4200 | 2.6200 | 2.6700 | 2.8500 | 3.0375 |
| R2D2 | 6.6100 | 3.9700 | 4.6900 | 5.1800 | 4.9800 | 3.9400 | 5.0600 | 4.7200 | 4.8938 |
| ALGD-ORB | 0.3460 | 0.4711 | 0.4207 | 0.3634 | 0.4051 | 0.5122 | 0.3835 | 0.4883 | 0.4237 |

features into grids using a quad-tree structure and selectively retaining the features with the highest response values within each grid. The goal is to achieve a more uniform distribution of extracted features. Uniform features can improve the pose calculation accuracy. The Mur-ORB algorithm effectively achieves the goal of uniformly distributing features by employing quad-trees for feature point management and optimization. In contrast, the ORB algorithm extracts feature points using a default threshold value of 30. However, the Mur-ORB algorithm enhances this process by utilizing a double threshold approach for more precise feature point extraction. It first uses a threshold of 20 to extract FAST keypoints, and if no points are extracted, the threshold is lowered to 7 for extraction. The Mur-ORB algorithm demonstrates the ability to extract feature points with a more uniform distribution, resulting in higher matching ac-curacy compared to the ORB algorithm. In the proposed ALGD-ORB algorithm presented in this paper, an adaptive FAST threshold is utilized to detect feature points during the feature point extraction stage. Unlike the fixed threshold used in Mur-ORB, the adaptive threshold in ALGD-ORB allows for the extraction of feature points that may have been missed by ORB in certain uniform regions. During the feature description stage, Mur-ORB improves the discriminability of feature descriptors by combining the gray size and feature descriptors generated using gray differences. However, it does not impose any limitations on the depth of the quad-tree, which can lead to excessive splitting and reduced computational efficiency. In contrast, the ALGD-ORB algorithm addresses this issue by adaptively restricting the quad-tree depth based on the aspect ratios of the input image. This adaptive approach allows for efficient termination of splitting, enhancing computational speed while maintaining accurate feature extraction. To evaluate the performance of the ALCD-ORB algorithm, other ORB-based and deep learning-based feature extraction methods were used. These algorithms are intended to provide a benchmark for accurate evaluation of the ALGD-ORB algorithm. The experiments demonstrate that enhancements to the ORB algorithm also face challenges regarding uneven feature point distribution, primarily due to the limitations of the feature point extraction algo-rithm itself. Therefore, improving the feature point extraction algorithm is the main method to improve the uniformity of feature point distribution, and further reasonable selection of the extracted feature points is needed to obtain more accurate feature point matching results. In the process of implementing feature point homogenization, attention should be paid to match-ing accuracy to avoid affecting the algorithm performance. Therefore, multiple factors need to be considered comprehensively when homogenizing feature points. Matching accuracy is not

solely determined by feature point extraction but is also closely tied to the feature description method. The discriminability of feature description vectors plays a vital role in achieving higher accuracy. Therefore, it is crucial to enhance the feature description methods to improve the homogenization process of feature points. To further improve the algorithm performance, feature points need to be uniformly distributed as much as possible while ensuring real-time performance and matching accuracy.

## Conclusions

This paper introduces a novel image feature extraction algorithm named Adaptive Threshold and Local Grayscale Difference-based ORB (ALGD-ORB). The ALGD-ORB algorithm is an enhanced version of the Mur-ORB algorithm, specifically designed to improve the feature point extraction and feature description stages. In the feature point detection stage, the algorithm employs an adaptive threshold technique, enabling the extraction of a higher number of feature points in uniform regions of the image. To mitigate the issue of excessive concentration and overlap of feature points, the quad-tree management method, as proposed by Mur-ORB, is employed. This approach ensures a more uniform dispersion of feature points throughout the image. To cope with the issue of the quad-tree being too slow, this paper sets different depths based on the feature points extracted from each pyramid image layer for different aspect ratios, effectively reducing the computation time. A binary string of grayscale difference information based on pixel patches is proposed in the feature description stage, integrating the grayscale size and the feature descriptor generated by grayscale difference information. Through verification on the Oxford dataset, the proposed ALGD-ORB algorithm effectively achieves the uniform distribution of feature points while achieving comparable performance to the ORB algorithm and its improvements.

## Acknowledgments

We sincerely appreciate the peer reviewers and editors of this article for their valuable suggestions and meticulous review, which have greatly enhanced the quality of this manuscript.

## Author Contributions

**Conceptualization:** Guoming Chu, Yan Peng, Xuhong Luo.

**Data curation:** Guoming Chu.

**Formal analysis:** Xuhong Luo.

**Funding acquisition:** Guoming Chu.

**Investigation:** Guoming Chu.

**Methodology:** Guoming Chu.

**Project administration:** Guoming Chu.

**Resources:** Guoming Chu.

**Software:** Guoming Chu, Yan Peng, Xuhong Luo.

**Supervision:** Guoming Chu.

**Validation:** Guoming Chu, Yan Peng.

**Visualization:** Guoming Chu.

**Writing – original draft:** Guoming Chu.

**Writing – review & editing:** Guoming Chu, Yan Peng, Xuhong Luo.

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
