## [Decision Letter · Decision Letter 0]

25 Aug 2023

PONE-D-23-24466Title - ALGD-ORB: An Improved Image Feature Extraction Algorithm with Adaptive Threshold and Local Gray DifferencePLOS ONE

Dear Dr. Chu,

Thank you for submitting your manuscript to PLOS ONE. After careful consideration, we feel that it has merit but does not fully meet PLOS ONE’s publication criteria as it currently stands. Therefore, we invite you to submit a revised version of the manuscript that addresses the points raised during the review process.

We look forward to receiving your revised manuscript.

Kind regards,

Muhammad Shahid Farid, Ph.D.

Academic Editor

PLOS ONE

Journal Requirements:

5. We note that Figure 5 in your submission contain [map/satellite] images which may be copyrighted. All PLOS content is published under the Creative Commons Attribution License (CC BY 4.0), which means that the manuscript, images, and Supporting Information files will be freely available online, and any third party is permitted to access, download, copy, distribute, and use these materials in any way, even commercially, with proper attribution. For these reasons, we cannot publish previously copyrighted maps or satellite images created using proprietary data, such as Google software (Google Maps, Street View, and Earth). For more information, see our copyright guidelines: http://journals.plos.org/plosone/s/licenses-and-copyright.

a. You may seek permission from the original copyright holder of Figure 5 to publish the content specifically under the CC BY 4.0 license.  

Reviewers' comments:

Reviewer's Responses to Questions

**Comments to the Author**

1. Is the manuscript technically sound, and do the data support the conclusions?

Reviewer #1: Yes

Reviewer #2: Partly

2. Has the statistical analysis been performed appropriately and rigorously? 

Reviewer #1: Yes

Reviewer #2: N/A

3. Have the authors made all data underlying the findings in their manuscript fully available?

Reviewer #1: Yes

Reviewer #2: Yes

4. Is the manuscript presented in an intelligible fashion and written in standard English?

Reviewer #1: Yes

Reviewer #2: Yes

5. Review Comments to the Author

Reviewer #1: In this paper, the authors have addressed an issue of dense and overlapping feature points, noise sensitivity, and imbalanced distribution, resulting in mismatches and redundancies in image feature points. An image feature extraction algorithm named ALGD-ORB is proposed to address this problem, utilizing adaptive threshold and local gray difference. Overall, this paper has a well-organized structure and is easy to follow. However, this manuscript still needs substantial revision, And I have the following concerns that need to be addressed:

1.The abstract should be abbreviated in a concise way.

2.The research motivation should be strengthened in the introduction. I recommend the authors to restructure the introduction into four subsections: background, current challenges or limits of prior SOTAs, research motivation, main contribution (here the authors may still use original text, but in a different order).

3. In related works，the authors should highlight what has been previously investigated or what has not been explored? On this basis, the novelty should be summarized by comparing with these listed literatures.

4.The confusion matrices should be provided.

5. What are units in Table 5? The computational complexity needs to be analyzed or compared.

6. I also recommend the authors to refer to several recent machine learning literatures.

a. 10.1109/TKDE.2023.3277839

b. 10.1109/JBHI.2022.3193148

c. 10.1016/j.knosys.2023.110789

d. 10.1109/TETCI.2021.3136642

Reviewer #2: In the paper ALGD-1 ORB: An Improved Image Feature Extraction Algorithm with Adaptive Threshold and Local Gray Difference is proposed an approach for image feature extraction, based on the ORB algorithm, that uses an adjustable threshold and a quadtree method, where different quadtree depths are set based on the total number of feature points extracted from the pyramid layers.

The subject addressed by the authors is of high interest, having applicability in different areas. Although, in my opinion, the presented approach be a proposed improvement of the ORB algorithm based on simple and well-known techniques, bringing little innovation, it could be a positive contribute, but there are several issues that need to be validated and supported in a more accurate way.

My comments are as follows.

Abstract - Authors should consider starting the abstract with a brief contextualization of the application area, and its importance, of the proposed work. Please, provide the full name of the ORB algorithm.

Some citations are not well formatted (for example: lines 33, 36 and 37).

From line 46 to line 100, the text seems to be the cover letter.

In the Introduction section, the authors present a somewhat confusing text, where some of their statements must be carefully reviewed, as they present some inaccuracies or lack foundation. For example:

Lines 100 to 102 - It is true that SIFT is a robust technique against scale changes and rotations. However, its effectiveness decreases in the presence of noise or when dealing with images with complex backgrounds.

Lines 107 to 110 - SURF overcome the SIFT limitation of operating in high-dimensional contexts, being also robust to lighting variations. But it is important to notice that this advantage entails a disadvantage compared to SIFT, since its performance in resource calculation is lower. Furthermore, SURF is not rotation invariant. Additionally, when the authors claim that the SURF algorithm allows real-time execution, it is important to substantiate because, as far as I know, the computational complexity of the SURF algorithm limits its application to real-time systems.

Line 130 - The text "... at the ICCV (IEEE International Conference on Computer Vision) in 2001, ..." is not needed.

Lines 134 to 135 - The sentence "It is about 100 times faster than SIFT algorithm and 10 times faster than SURF algorithm." must be sustained in references.

Lines 225 to 227 - Please, revise the sentence "Mur-Arta [20,21] employed a fixed double-threshold FAST method in the initial..."

Line 253 - When using the acronym oFAST for the first time, the name should be written in full.

Line 254 - When using the acronym oFAST for the first time, the name should be written in full.

Equation 10 - The equation represents the formula for calculating the threshold value for each grid of the image, so that I_i_max, I_i_min and the largest/smaller gray levels will be the values relative to each grid of the image and not to the whole image.

Line 462 - Please, correct the typo.

Lines 714 to 716 - Considering the values in table 5, I do not understand the statement "These results highlight the outstanding runtime efficiency of the ALGD-ORB algorithm on the Oxford dataset, making it a promising choice, particularly in resource-constrained scenarios."

Table 5 - What units of measurement are used?

The authors compare the proposed algorithm with methods proposed some time ago. There are some recent algorithms where the focus is on improving the ORB algorithm, with which I think it would be positive to compare the work of the authors. This comparative study with more recently proposed approaches would help to understand more clearly the current contribution of the proposed work, as well as its possible innovation.

Most references are old. As far as I could see, there is only one reference from 2023, with almost all the rest being earlier than 2020. Authors should consider more recent work.

6. PLOS authors have the option to publish the peer review history of their article (what does this mean?). If published, this will include your full peer review and any attached files.

Reviewer #1: No

Reviewer #2: No

---

## [Author Response · Author response to Decision Letter 0]

21 Sep 2023

Response to Editor and Reviewers

Dear Editor and Reviewers,

Thank you for taking the time to review my manuscript. I have carefully considered your comments and suggestions.

Response to Editor:

I have updated my responses to your comments in the cover letter, which is included in the revised submission package.

Response to Reviewers:

Detailed responses to the reviewers' comments have been provided in a separate file labeled 'Response to Reviewers'.

I appreciate your time and effort in reviewing my work, and I look forward to your feedback on the revised manuscript.

Best regards,

Guoming Chu

School of Automation and Information Engineering,

Sichuan University of Science and Engineering

Email: 759053364@qq.com

---

## [Decision Letter · Decision Letter 1]

6 Oct 2023

ALGD-ORB: An Improved Image Feature Extraction Algorithm with Adaptive Threshold and Local Gray Difference

PONE-D-23-24466R1

Dear Dr. Chu,

We’re pleased to inform you that your manuscript has been judged scientifically suitable for publication and will be formally accepted for publication once it meets all outstanding technical requirements.

Kind regards,

Muhammad Shahid Farid, Ph.D.

Academic Editor

PLOS ONE

Reviewers' comments:

Reviewer's Responses to Questions

**Comments to the Author**

1. If the authors have adequately addressed your comments raised in a previous round of review and you feel that this manuscript is now acceptable for publication, you may indicate that here to bypass the “Comments to the Author” section, enter your conflict of interest statement in the “Confidential to Editor” section, and submit your "Accept" recommendation.

Reviewer #1: All comments have been addressed

Reviewer #2: All comments have been addressed

2. Is the manuscript technically sound, and do the data support the conclusions?

Reviewer #1: Yes

Reviewer #2: (No Response)

3. Has the statistical analysis been performed appropriately and rigorously? 

Reviewer #1: Yes

Reviewer #2: (No Response)

4. Have the authors made all data underlying the findings in their manuscript fully available?

Reviewer #1: Yes

Reviewer #2: (No Response)

5. Is the manuscript presented in an intelligible fashion and written in standard English?

Reviewer #1: Yes

Reviewer #2: (No Response)

6. Review Comments to the Author

Reviewer #1: The authors have addressed my concerns well, and I can recommend this paper for a publication in this journal.

Reviewer #2: (No Response)

7. PLOS authors have the option to publish the peer review history of their article (what does this mean?). If published, this will include your full peer review and any attached files.

Reviewer #1: No

Reviewer #2: No

---

## [Editor Report · Acceptance letter]

11 Oct 2023

PONE-D-23-24466R1 

ALGD-ORB: An Improved Image Feature Extraction Algorithm with Adaptive Threshold and Local Gray Difference 

Dear Dr. Chu:

I'm pleased to inform you that your manuscript has been deemed suitable for publication in PLOS ONE. Congratulations! Your manuscript is now with our production department. 

Kind regards, 

on behalf of

Dr. Muhammad Shahid Farid 

Academic Editor

PLOS ONE